# Towards Efficient Unroll Generalization in Learned Optimizers

## Abstract

Recent works have demonstrated that learned optimizers (LOs) can be competitive and sometimes even outperform hand-designed counterparts, highlighting their potential as a pathway toward developing better optimization algorithms. Yet, despite this promise, meta-generalization remains a major challenge for LOs. In particular, they often struggle to maintain stable convergence over long unrolls, as they are typically meta-trained only on short horizons. While extending the unroll length during meta-training may seem like a natural remedy, in practice it substantially increases computational cost (at least linearly) and frequently leads to divergence or collapse due to compounding errors. To improve the long unroll generalization of LOs, we propose a novel meta-training scheme called Efficient Long-horizon Learning (ELO), which leverages a replay buffer to efficiently extend unroll length during meta-training without adding extra meta-training cost. In addition, it integrates online behavior cloning to stabilize meta-training and potentially inherit the generalization benefits of hand-designed optimizers. We evaluate ELO on a variety of vision and language tasks, showing its success in achieving long-unroll generalization in practical scenarios.

## 1 Introduction

The remarkable achievements of deep neural networks have been closely tied to the evolution of optimization algorithms (Sun et al., 2019; Sun, 2020; Abdulkadirov et al., 2023). Learned Optimizers (LOs), as a rising paradigm, have demonstrated the ability to discover superior update rules(Andrychowicz et al., 2016; Metz et al.; Chen et al., 2020; Thérien et al., 2024), achieving faster and better convergence than hand-designed optimizers on certain tasks. Despite their potential, LOs are still far from mature (Wichrowska et al., 2017). A key challenge is that they often saturate quickly or even gradually diverge when evaluated on very long unrolls in downstream tasks (e.g., $10 \times$ larger than the maximum unroll length used in meta-training), which is critical for practical model training.

In this work, we focus on improving the unroll generalization of LOs in an efficient way. We observe that common meta-training setups not only struggle to generalize to long unrolls, but also inadvertently waste training resources. To address this, we introduce an efficient long-horizon learning paradigm (ELO), which effectively "recycles" computational resources and reallocating them toward long-unroll meta-training. ELO integrates a replay buffer(Ross et al., 2011), enabling LOs to experience very long unrolls during meta-training without incurring extra computational cost, thereby equipping them with stronger long-horizon generalization. However, naively applying a replay buffer destabilizes meta-training in its early stages due to compounding errors (Ross et al., 2011; Ross & Bagnell, 2014). To address this, we further incorporate behavior cloning(Rajaraman et al., 2020; Torabi et al., 2018), guiding the LO to imitate a hand-designed optimizer (Adam, in our case) while still allowing it to improve beyond the teacher.

Our main contributions are as follows:

- We introduce a replay buffer mechanism that enables LOs to observe sufficiently long unrolls during meta-training without additional computational overhead.
- We propose a behavior cloning strategy to stabilize early meta-training and to transfer potential generalization benefits from an *expert* optimizer.

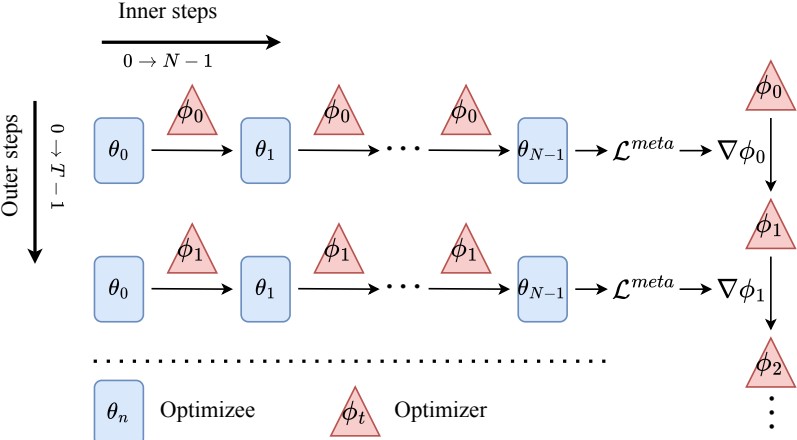

Figure 1: Overview of the meta-training of a LO. At each outer step $t$, the optimizer (LO) parameters $\phi_t$ are used to update the optimizee parameters $\theta_n$ over a sequence of $N$ inner steps. During this unroll, at every inner step $n$, a meta-loss $\mathcal{L}_n^{meta}$ is computed based on the optimizee's performance. These per-step losses are accumulated to form $\mathcal{L}^{meta}$. The resulting meta-gradient $\nabla\phi_t$ is then used to update the optimizer parameters, producing $\phi_{t+1}$. This iterative process enables the LO to improve its update rules across successive outer steps.

- We empirically demonstrate that, across both vision and language tasks, ELO consistently outperforms strong hand-designed and learned baselines.

## 2 RELATED WORK

**Learned Optimizers (LOs).** LOs employ trainable models (e.g., MLP) to replace hand-crafted optimization algorithms (Andrychowicz et al., 2016). Previous literature has proposed a variety of approaches to improve LOs. Metz et al.; 2022) explored new architectures and large-scale training regimes for LOs. Yang et al. (2021); Thérien et al. (2024) introduced techniques such as maximal update parameterization to leverage LOs in the training of large-scale models. (Chen et al., 2020) explored applying imitation learning to LOs, typically in an off-policy manner to build stronger baselines. But such approaches inevitably suffer from compounding error due to their alternating offline training.

**Replay Buffers.** Replay buffers are widely used in reinforcement learning (RL)(Ross et al., 2011; Liu & Zou, 2018; Zhang & Sutton, 2017) and continual learning (CL)(Rolnick et al., 2019; Chaudhry et al., 2021). They act as a core mechanism to store trajectories or transitions collected during training, enabling the learner to sample from past experiences rather than relying solely on the most recent data. In RL, this helps break temporal correlations(Mnih et al., 2015), improve sample efficiency(Schaul et al., 2015), and stabilize learning, while in CL it plays a key role in alleviating catastrophic forgetting(Rolnick et al., 2019; Buzzega et al., 2020). Instead of leveraging its conventional replay benefits, we use replay buffer in our work to enable efficient sampling for long unroll meta-training.

**Behavior Cloning.** Behavior Cloning (BC)(Rajaraman et al., 2020; Torabi et al., 2018) is one of the most widely used paradigms in imitation learning(Pomerleau, 1988; Osa et al., 2018; Liu et al., 2021), where the objective is to approximate an expert policy by directly regressing from observed states to expert actions. Formally, given a dataset of expert demonstrations $\mathcal{D} = \{(x_i, a_i^\star)\}_{i=1}^N$, where $x_i \in \mathcal{X}$ denotes the state and $a_i^\star \in \mathcal{A}$ is the corresponding expert action, the goal is to learn a policy $\pi_\theta : \mathcal{X} \to \mathcal{A}$, parameterized by $\theta$, that closely approximates the expert's behavior. A simple

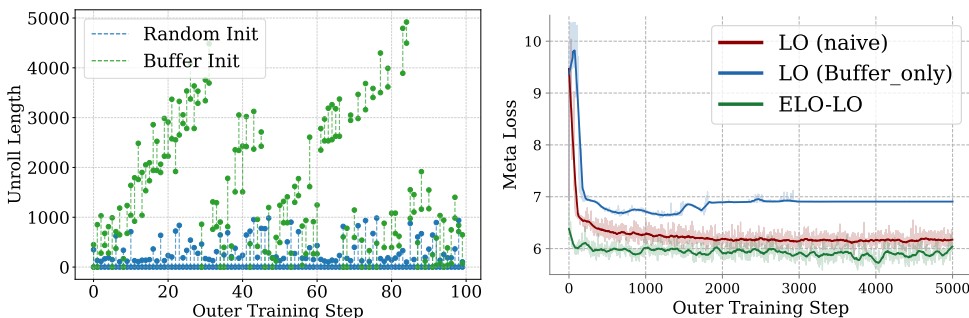

Figure 2: **Left:** Buffer-based initialization allows the unroll length to grow more efficiently than random initialization. **Right:** ELO ensures the stability of meta-training when LO encounter very long unrolls.

version of behavior cloning solves the supervised regression problem(Florence et al., 2022):

$$\hat{\pi}_\theta = \arg\min \pi_\theta \; \frac{1}{N} \sum_{i=1}^{N} \|\pi_\theta(x_i) - a_i^\star\|_2^2. \tag{1}$$

In our work, we leverages online behavior cloning primarily as a stabilizer for buffer-based meta-training, enabling efficient long-unroll meta-training.

## 3    OVERVIEW OF LEARNED OPTIMIZERS

In our work, we adopt the `small_fc_lopt` architecture (Metz et al., 2022) as LO, a three-layer MLP with ReLU activations(Nair & Hinton, 2010). The optimizer takes as input a feature vector (e.g. gradient, momentum, rms, etc.) for each parameter in the optimizee and outputs an update direction, $d$, and magnitude, m. That is, $f_\phi(\cdot) = [d, m]$, where $f$ is the learned optimizer and $\phi$ are its parameters. The optimizee's parameters $\theta$ are then updated as follows:

$$\theta_t = \theta_{t-1} - \lambda_1 d \times e^{\lambda_2 m}, \tag{2}$$

where $\lambda_1$ and $\lambda_2$ are hyper-parameters. In general, learning the meta-parameters, $\phi$, involves solving an optimization problem of the form (Thérien et al., 2024):

$$\min_\phi \; \mathbb{E}_{\tau \sim \mathcal{D}} \Big[ \sum_{n=1}^{N} \mathcal{L}_n^{\text{meta}}(\theta_n; \phi) \Big], \tag{3}$$

where $\mathcal{D}$ is a distribution of tasks. The objective seeks to minimize the sum of per-timestep losses over the training horizon $N$. General pipeline of how a LO is trained is shown in **Figure** 1.

## 4    EFFICIENT LONG-HORIZON LEARNING

In this section, we first analyze the limitations of commonly adopted meta-training frameworks in terms of unroll generalization and computational efficiency. To address these issues, we introduce Efficient Long-horizon Learning (ELO), which leverages a replay buffer to extend unroll lengths observed during meta-training without incurring additional computational cost. To further stabilize buffer-based training and improve overall generalization, we incorporate an online behavior cloning strategy. Details are provided in **Algorithm** 1.

### 4.1    UNROLL INITIALIZATION FROM REPLAY BUFFER FOR LOS

**Random Initialization.**    In existing meta-learning setups for learned optimizers, every unroll is randomly initialized(Metz et al., 2019; Thérien et al., 2024), with the unroll length $N$ sampled from a log-uniform distribution: $p(N) = \frac{1}{N \log \frac{N_{\max}}{N_{\min}}}, \quad N \in (N_{\min}, N_{\max}]$, an example is provided in **Figure 2**. In practice, $N_{\min}$ and $N_{\max}$ are usually set to be small (e.g. $N_{\min} = 100$ and

---

**Algorithm 1:** Efficient Long-horizon Learning (ELO)

---

**Input:** The size $M$ of replay buffer, the threshold $P_{\text{th}}$ of applying buffer initialization.

**Initialize:** Replay buffer $\mathcal{B} = \{s_1, s_2, \ldots, s_m\}, m \leq M$, where $s_*$ indicates a inner state.

**for** $t = 0, 1, 2, ..., T - 1$ **do**

  Sample $P_{\mathcal{B}} \sim \text{Uniform}(0, 1)$;

  **if** $(P_{\mathcal{B}} > P_{\text{th}}) \wedge (t > 0)$ **then**

  |   Randomly select $s_*$ from $\mathcal{B}$ $\left(K := \pi_\zeta(s_*)\right)$

  **else**

  |   Randomly initialize inner state ($K = 0$)

  **end**

  sample $N_{\text{push}} \in [K, N + K) \cap \mathbb{Z}$;

  **for** $n = K, K + 1, K + 2, ..., K + N - 1$ **do**

  |   $\theta_{n+1}^{\mathcal{H}} = \theta_n + \Delta\theta_n^{\mathcal{H}}$;

  |   $\theta_{n+1}^{\mathcal{O}} = \theta_n + \Delta\theta_n^{\mathcal{O}}$;

  |   $\theta_{n+1} = (1 - \alpha_t)\theta_{n+1}^{\mathcal{H}} + \alpha_t\theta_{n+1}^{\mathcal{O}}$;

  |   $\mathcal{L}_n^{meta} = (1 - \alpha_t)\mathcal{L}_n^{bc}(\theta_n^{\mathcal{H}}, \theta_n^{\mathcal{O}}) + \alpha_t \mathcal{L}_n^{task}(\theta_n; \phi, \alpha_t)$;

  |   $s_{n+1} = (\theta_{n+1}, \zeta_{n+1})$;

  |   **if** $n == N_{push}$ **then**

  |   |   $\mathcal{B} \leftarrow \begin{cases} \text{enqueue}(\mathcal{B}, s_n), & m < M \\ \text{enqueue}(\text{dequeue}(\mathcal{B}), s_n), & m = M \end{cases}$;

  |   **end**

  **end**

  $g_t = \mathcal{G}\left(\sum_{n=K+1}^{K+N} \mathcal{L}_n^{meta}(\theta_n^{\mathcal{H}}, \theta_n^{\mathcal{O}}; \phi_t, \alpha_t)\right)$;

  $\phi_{t+1} = \mathcal{U}(g_t, t; \phi_t)$

**end**

---

**Notation:**

we highlight buffer-related operations in green and behavior cloning (BC)-related components in yellow for clarity.

$\mathcal{H}$: Adam
$\mathcal{O}$: learned optimizer
$\theta$: parameters of optimizee
$\phi$: parameters of LO
$\zeta$: auxiliary accumulators (e.g. momentum)
$\pi_\zeta(s)$: projection operator extracting the step index from state $s$
$P_{\mathcal{B}}$: probability of buffer init
$K$: start step of inner loop
$N_{\text{push}}$: inner step to push into $\mathcal{B}$
$\mathcal{G}$: meta-gradient estimator, we use persistent evolution strategies (PES)
$\mathcal{U}$: update operator

---

$N_{\max} = 1,000$), to mitigate compounding errors. The expected number of times that an inner step $n$ contributes to the meta-gradient across training is proportional to

$$\Pr[N > n] = \int_n^{N_{\max}} \frac{1}{N \, \log \frac{N_{\max}}{N_{\min}}} \, dN = \frac{\log \frac{N_{\max}}{n}}{\log \frac{N_{\max}}{N_{\min}}}. \tag{4}$$

For early steps, $\Pr[N > n] \to 1$ as $n \to N_{min}$, and for later steps, $\Pr[N > n] \to 0$ as $n \to N_{max}$. As a result, the LO prefers to optimize the early training regime, and melts down on long unroll optimization.

A better way is to gradually increase $N_{max}$ to include longer unrolls during meta-training. However, since the unroll length of downstream tasks is task-agnostic, $N_{\max}$ typically has to grow quite large to cover most cases. Because meta-training for LOs is typically very expensive, and its computational cost scales almost linearly with the unroll length, this makes it impractical for real-world applications.

**Buffer Initialization.** In the above schemes, a substantial amount of compute were wastes on early unroll training. Because the learned optimizer (LO) constantly revisits the initial inner steps at each unroll, even after it has already become proficient at optimizing them. To improve this, we propose to maintain a replay buffer that stores intermediate checkpoints (inner states) from ongoing unrolls. When initializing a new unroll, instead of always restarting from a random initialization, the LO has a probability of being initialized from one of the buffered checkpoints. This doesn't add additional training cost, but shifts computation away from repeatedly revisiting the earliest trajectory segments and reallocates it toward training on longer effective unrolls. An example of unroll sampling using buffer initialization is shown in **Figure 2**.

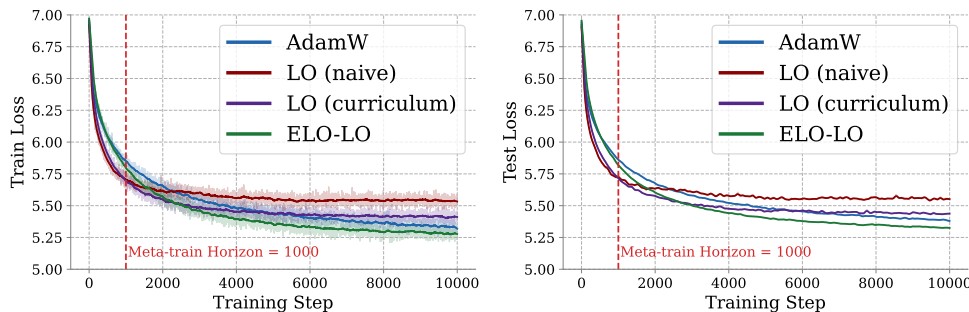

Figure 3: In-distribution (ID) evaluation on ImageNet-1K (32×32) using a 3-layer MLP with hidden width 128. We report both training loss (Left) and validation loss (Right) across $10,000$ steps. The red dashed line indicates the meta-training horizon ($N = 1000$).

Specifically, we design the buffer $\mathcal{B}$ as a queue of default size $|\mathcal{B}| = 4$. Let $P_{\text{th}} \in (0, 1)$ be a threshold parameter. At the beginning of each unroll, we draw $P_{\mathcal{B}} \sim \text{Uniform}(0, 1)$. If $P_{\mathcal{B}} > P_{\text{th}}$, the new unroll is initialized from a randomly chosen checkpoint in $\mathcal{B}$; otherwise, it is initialized from scratch. To further increase the probability of sampling very long unrolls, we replace log-uniform sampling with uniform sampling for the unroll length. By decreasing the threshold $P_{\text{th}}$, the expected unroll length $\mathbb{E}[N]$ of the mixture distribution shifts upwards, thereby increasing $\Pr[N > n_{large}]$. This encourages the LO to dedicate more attention on long unroll optimization learning. Besides, keeping $P_{\text{th}} > 0$ guarantees that the LO can occasionally review the initial regime. Detailed descriptions of how buffer works are highlight in yellow in **Algorithm 1**.

### 4.2 Composition of Optimization Trajectories

Our goal is to expose the LO to sufficiently long unrolls across meta-training, which motivates setting a small threshold $P_{\text{th}}$. However, this likely results in generating large unroll length $N$ in the early stages of meta-training, which can be problematic, as shown in **Figure 2**. At this stage, the LO remains underfit and tends to produce suboptimal optimization trajectories. Formally, let the optimizee trajectory $\mathcal{T}$ be defined as $\mathcal{T} = \{(\theta_n, \Delta\theta_n)\}_{n=0}^N$, where

$$\theta_{n+1} = \theta_n + \Delta\theta_n, \quad n = 0, \dots, N-1, \tag{5}$$

and

$$\Delta\theta_n = f(\Delta\theta_{<n}; \theta_0, \phi). \tag{6}$$

Approximation errors $\epsilon_n$ accumulate over time, which yields

$$\|\theta_N - \theta_N^\star\| \leq \sum_{n=1}^N \kappa^{N-n} \|\epsilon_n\|, \tag{7}$$

where $\kappa$ is the Lipschitz constant (Bubeck et al., 2015) of the update dynamics, typically large in early meta-training. As $N$ grows, the accumulated error $\|\theta_N - \theta_N^\star\|$ can amplify exponentially, a phenomenon commonly referred to as compounding error. As a result, the task-driven meta-gradients from the last inner step $\nabla_\phi \mathcal{L}_N^{task}(\theta_{N-1}; \phi)$ degenerates into pure noise, and the cumulative meta-gradients $\sum_{n=1}^N \nabla_\phi \mathcal{L}_n^{task}(\theta_{n-1}; \phi)$ becomes highly uninformative, making the training hard to progress.

To stabilize meta-training, we propose to incorporate hand-designed optimizers (we use Adam in this paper) to improve the trajectory, given their well-known ability to generate stable and high-quality optimization paths. Specifically, for any inner step $n$, we combine the trajectory produced by Adam and the one produced by the LO in the following way:

$$\begin{aligned} \theta_{n+1}^{\mathcal{H}} &= \theta_n + \Delta\theta_n^{\mathcal{H}}, \\ \theta_{n+1}^{\mathcal{O}} &= \theta_n + \Delta\theta_n^{\mathcal{O}}, \\ \theta_{n+1} &= (1 - \alpha_t)\,\theta_{n+1}^{\mathcal{H}} + \alpha_t \theta_{n+1}^{\mathcal{O}}, \end{aligned} \tag{8}$$

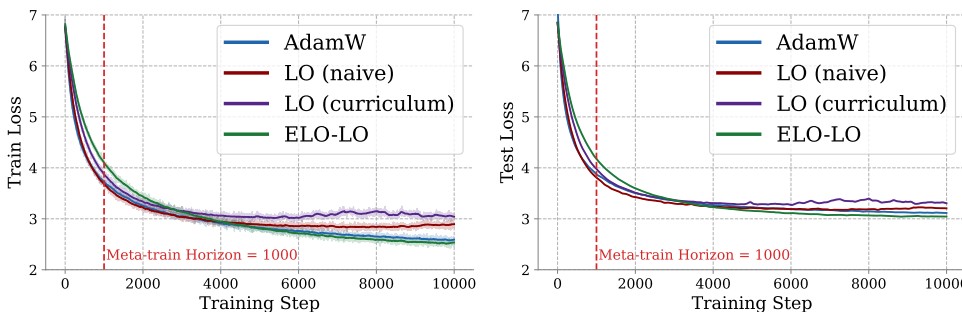

Figure 4: Out-of-distribution (OOD) evaluation on ImageNet-1K (32×32) using a ResNet18. We report both training loss (Left) and validation loss (Right) across $10,000$ steps. The red dashed line indicates the meta-training horizon ($N = 1000$).

where $\mathcal{H}$ and $\mathcal{O}$ denote Adam and LO, respectively, and $\alpha_t$ gradually increases from $0$ to $1$ as the outer step $t$ grows ($\alpha_t = \frac{t}{T-1}$ ($t \in \mathbb{Z}$, $0 \leq t < T$)). This would effectively reduce $\kappa$ to a small value in the early phase of meta-training, as $\|\theta_N^{\mathcal{H}} - \theta_N^{\star}\| \ll \|\theta_N^{\mathcal{O}} - \theta_N^{\star}\|$ at this stage. As the meta-training progresses, the quality of the trajectories generated by the LO are expected to gradually improve. Accordingly, the weighting gradually shifts from relying entirely on Adam to relying fully on the LO. This adaptive transition ensures that every unroll trajectory during meta-training remains of high quality, thereby avoiding redundant meta-training and instability that noisy trajectories would otherwise introduce.

### 4.3 ONLINE BEHAVIOR CLONING

For any inner step $n$, directly relying on the fused trajectories makes the task-driven meta-loss $\mathcal{L}_n^{task}(\theta_n; \phi, \alpha_t)$ less informative to LO during early stage meta-training, as $\theta_n$ is dominant by $\theta_n^{\mathcal{H}}$ at this time. We then leverage a regularization loss $\mathcal{L}_n^{bc}(\theta_n^{\mathcal{H}}, \theta_n^{\mathcal{O}})$ to guide the LO at each inner step $n$, replacing the noisy $\mathcal{L}_n^{task}(\theta_n; \phi, \alpha_t)$ with a more accurate signal. $\mathcal{L}_n^{bc}(\theta_n^{\mathcal{H}}, \theta_n^{\mathcal{O}})$ is defined as:

$$\mathcal{L}_n^{bc}(\theta_n^{\mathcal{H}}, \theta_n^{\mathcal{O}}) = \sum_{n=1}^{N} \|\theta_n^{\mathcal{O}} - \theta_n^{\mathcal{H}}\|_2^2. \tag{9}$$

While training the LO solely under $\mathcal{L}^{bc}n(\theta n^{\mathcal{H}}, \theta_n^{\mathcal{O}})$ ensures stability, its performance will be inherently bounded by that of Adam. Our ultimate goal, by contrast, is to train an LO capable of surpassing hand-designed optimizers on specific tasks. To this end, we further propose to combine $\mathcal{L}_n^{bc}(\theta_n^{\mathcal{H}}, \theta_n^{\mathcal{O}})$ with $\mathcal{L}_n^{task}(\theta_n; \phi, \alpha_t)$ through a convex combination:

$$\mathcal{L}_n^{meta}(\theta_n^{\mathcal{H}}, \theta_n^{\mathcal{O}}; \phi, \alpha_t) = (1 - \alpha_t)\,\mathcal{L}_n^{bc}(\theta_n^{\mathcal{H}}, \theta_n^{\mathcal{O}}) + \alpha_t\,\mathcal{L}_n^{task}(\theta_n; \phi, \alpha_t), \tag{10}$$

where $\alpha_t$ is set the same as in **Eq.** 8. In this way, the meta-gradients are initially dominated by $\mathcal{L}_n^{bc}(\theta_n^{\mathcal{H}}, \theta_n^{\mathcal{O}})$, ensuring stable training signals in the early meta-training stage, and potentially allowing the LO to absorb generalization benefits from Adam. At this stage, it also mitigates noisy meta-gradients that would otherwise arise from the underfitting of the LO, thereby further accelerating its convergence. Over time, the meta-gradients gradually shift towards being fully driven by $\mathcal{L}_n^{task}(\theta_n; \phi, \alpha_t)$, encouraging the LO to discover superior optimization rules. Illustrative uses of trajectory fusion and *expert* forcing are highlighted in yellow in **Algorithm 1**.

## 5 EMPIRICAL EVALUATION

In this section, we evaluate ELO through extensive experiments across vision and language domains, using various datasets and architectures. We empirically show:

- How buffer initialization achieves efficient long unroll sampling, and the effect of setting different buffer threshold $P_{\text{th}}$.

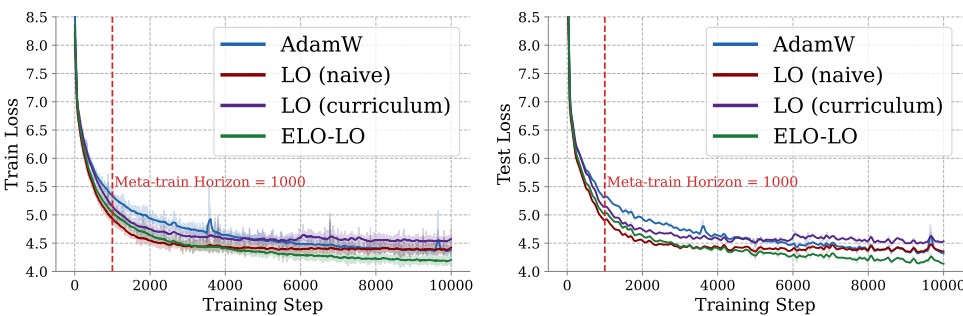

Figure 5: Out-of-distribution (OOD) evaluation on FineWeb 10B using gpt2-mini. We report both training loss (Left) and validation loss (Right) across 10,000 steps. The red dashed line indicates the meta-training horizon ($N = 1000$).

- Across in distribution (ID) and out of distribution (OOD) tasks, ELO constantly surpasses other baselines in very long unroll regime.
- We ablate each component of ELO to assess its role and interactions with the others.

## 5.1 EXPERIMENTAL SETUP

**Meta-training.** Our experimental setup and meta-training pipeline largely follow that of (Metz et al., 2022; Thérien et al., 2024), with best hyper-parameters applied for each baseline. Specifically, our optimzers follow the small_fc_lopt architecture and are 3-layer MLPs with a hidden width of 32 as LOs, which take a variety of input features (e.g. gradient, momentum, rms, etc.) inspired by (Maheswaranathan et al., 2021). For all LO baselines, we meta-train using AdamW (Loshchilov & Hutter, 2017) with an initial learning rate of $3 \times 10^{-3}$. The step_mult $\lambda_1$ and exp_mult $\lambda_2$ mentioned in **Eq. 2** are both set to 0.001 by default. For ELO-LO, the *expert* is set to Adam(Kingma & Ba, 2014) with learning rate $1 \times 10^{-3}$, although we also experimented with other hand-designed optimizers (e.g. AdamW) as the expert but did not yet observe improvements beyond Adam. The meta-trainings are performed on ImageNet-1K (Deng et al., 2009), resized to $32 \times 32$, using a batch size of 4096, for 5,000 outer steps with an maximum unroll length of 1,000. For LO (curriculum), the maximum unroll length grows from 1,000 to 10,000 as outer step t proceeds. Standard data augmentations such as random flipping, cropping, and translation, are applied during meta-training to avoid overfitting(Krizhevsky et al., 2012; Cubuk et al., 2018). Instead of using standard ES algorithm, we estimate meta-gradients using persistent evolution strategies (PES) (Vicol et al., 2021) for faster training, with a truncation length of 50.

**Meta-testing.** We conduct evaluation across various vision and language tasks using different models. For vision tasks, we evaluate on ImageNet-1K (32×32 resolution) using two representative architectures: a 3-layer MLP with hidden width 128, and ResNet-18 (He et al., 2016a;b). Both models are trained with a batch size of 4096, using the same augmentations as in meta-training. For langue tasks, we employed the popular FineWeb-10B (Penedo et al., 2024) dataset along with the GPT-2-mini (Radford et al., 2019) model. Due to computational constraints, the batch size for language experiments is set to 512. For all AdamW (Loshchilov & Hutter, 2017) baselines, we set the weight decay to be $1 \times 10^{-4}$, and search the best learning rate from [0.01, 0.007, 0.004, 0.001, 0.0007, 0.0004, 0.0001] for each task.

## 5.2 REPLAY BUFFER VS. BEHAVIOR CLONING

We begin by providing a brief analysis of the meta-training process.

**Sampling efficiency.** As shown in **Figure** 2 (left), although buffer initialization requires exactly the same computational cost as random initialization (evidenced by the equal lengths of the two line segments in each column), it significantly increases the effective unroll length $N$. In this case, we set the buffer threshold $P_{th} = 0.1$; lowering $P_{th}$ further increases the expected unroll length, whereas raising it has the opposite effect.

**Meta-training stability.** However, using the buffer alone leads to severe compounding errors during meta-training, which can cause the training process to become unstable or even collapse. As shown in **Figure** 2 (right), training with buffer-only initialization causes the final meta-loss to converge toward $\approx 6.91$ (corresponding to the entropy of a uniform random output over 1,000 classes, i.e., $\log 1000$), reducing the LO to random guessing(Goodfellow et al., 2016). In contrast, when behavior cloning is additionally applied, the meta-loss remains consistently stable throughout meta-training, demonstrating the effectiveness of this combination.

## 5.3 EFFECT OF BUFFER THRESHOLD $P_{\text{th}}$

We use $P_{\text{th}}$ to control both the expected and maximum lengths of the sampled unrolls. To better understand how downstream performance varies with $P_{\text{th}}$, we conduct experiments on ImageNet-1K (32×32) using an MLP with hidden width 128. Specifically, we meta-train ELO-LO under different values of $P_{\text{th}}$ (0.01, 0.05, 0.1, 0.2, 0.3, 0.5, 0.7) and evaluate their performance. As shown in **Figure 6**, the best results are achieved when $P_{\text{th}} = 0.2$, while either decreasing or increasing $P_{\text{th}}$ leads to degraded performance.

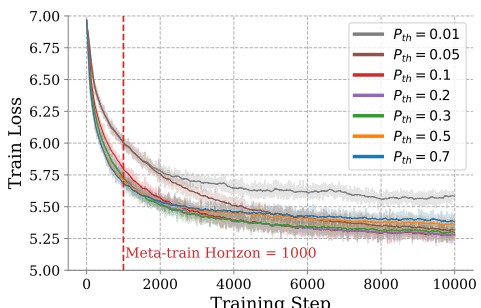

Figure 6: Effect of buffer threshold $P_{\text{th}}$ on downstream performance. The best performance occurs at $P_{\text{th}} = 0.2$, while both lower and higher thresholds degrade generalization due to insufficient coverage of short or long unrolls.

We hypothesize that this phenomenon arises because reducing $P_{\text{th}}$ from 0.7 to 0.2 appropriately reallocates computational resources toward longer unrolls, thereby enhancing long-horizon learning. However, further decreasing $P_{\text{th}}$ causes insufficient training on short unrolls, which in turn weakens the optimizer's ability to handle both short and long unrolls effectively. In subsequent comparative experiments, we therefore adopt the ELO-LO meta-trained with the best-performing threshold for all tasks.

## 5.4 EVALUATING GENERALIZATION TO LONG UNROLLS

We evaluate the long-horizon generalization ability of ELO across both vision and language tasks. During meta-training, the maximum unroll length sampled is limited to $1,000$, while evaluation is extended to $10,000$-step unrolls to assess performance far beyond the meta-training horizon. Experiments cover multiple model architectures, including MLPs and ResNet-18 on ImageNet-1K (32×32), as well as GPT2-mini on the FineWeb-10B dataset.

As shown in **Figures** 3, 4, and 5, ELO-LO consistently achieves the lowest final training loss across all tasks, outperforming strong baselines such as AdamW, naive-LO, and curriculum-LO. While AdamW maintains stable convergence but plateaus at higher loss values, naive-LO quickly flattens and even diverges beyond the training horizon. By contrast, ELO-LO matches AdamW's stability while continuing to reduce loss on long unrolls.

To further validate generalization, we also evaluate on held-out test sets. As summarized in **Table** 1, ELO-LO consistently achieves the highest accuracy and the lowest test loss across all benchmarks, confirming that the method provides stable improvements in both in-distribution and out-of-distribution evaluation.

## 5.5 ABLATING ELO'S COMPONENTS

We further disentangle the contributions of ELO's core components through a systematic ablation study. Specifically, we evaluate (i) a naive LO meta-trained with random initialization, (ii) a variant with uniform unroll sampling, (iii) the addition of behavior cloning (BC), and (iv) the complete LoL-LO framework that integrates both BC and the replay buffer. We further disentangle the contributions of each component in ELO through ablation studies.

Table 1: Best performance of different methods on vision and language tasks. For vision tasks (ImageNet-1K (32×32)), we report test accuracy (%). For language tasks (FineWeb-10B), we report test cross entropy loss.

| Method | ImageNet-1K MLP | ImageNet-1K ResNet18 | FineWeb-10B GPT2-mini |
|---|---|---|---|
| Metric | Acc. (%) | Acc. (%) | CE-Loss |
| AdamW | 8.26 | 35.62 | 4.31 |
| LO (naive) | 6.69 | 33.83 | 4.29 |
| LO (curriculum) | 7.77 | 31.96 | 4.44 |
| ELO-LO | **8.77** | **36.51** | **4.13** |

As shown in **Figure** 7, the naive LO quickly saturates and fails to improve beyond a moderate level of unroll generalization. Incorporating uniform unroll sampling alleviates this issue by making the training trajectories sparser. Adding BC enables the LO to inherit the strong inductive bias of Adam, leading to a noticeable improvement in unroll generalization even without buffer support. Finally, combining behavior cloning with the replay buffer yields the complete LoL-LO, which achieves the strongest long-horizon generalization. Note that we do not report a "buffer-only" variant, as introduced in subsection 5.2, using buffer initialization without behavior cloning often causes meta-training to become unstable or even collapse.

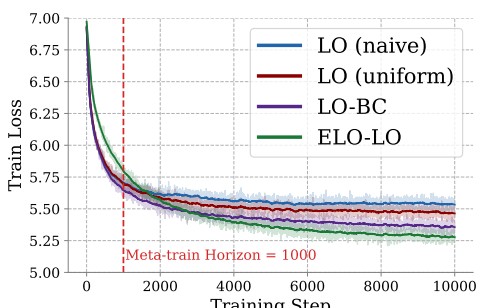

Figure 7: Ablation study on ImageNet-1K (32×32) with a 3-layer MLP (width 128). We showed how each component contributes to long unroll generalization.

# 6 DISCUSSION.

We have validated that ELO brings clear improvements for long unroll optimization across various tasks. In this work, we primarily used Adam as the expert. We also made some preliminary attempts with more advanced hand-designed optimizers such as AdamW (Loshchilov & Hutter, 2017) and Muon (Jordan et al., 2024). However, we found it challenging to make these alternatives work reliably in practice, and further effort is still required in this direction. Another important limitation lies in scale. Because meta-training LOs is extremely resource-intensive, our experiments have been limited to smaller models and datasets. As a result, the LOs trained still struggle to generalize effectively to optimizing very large-scale architectures, a capability that is essential for practical deployment. Despite this, ELO provides a promising step forward by effectively improving the efficiency and stability of meta-training. Looking ahead, we plan to extend our exploration to large-scale datasets and models, paving the way for LOs trained with ELO to serve as practical alternatives to state-of-the-art hand-designed optimizers in real-world systems.

# 7 CONCLUSION.

In this work, we introduced ELO: An efficient long horizon meta-training scheme for learned optimization. By leveraging replay buffers with expert guidance from online behavior cloning, ELO stabilizes meta-training, allowing the optimizer to learn from very long unrolls from the beginning of training. Our experiments across vision and language benchmarks demonstrate that learned optimizers meta-learned with ELO consistently outperform strong baselines, highlighting ELO's potential as a practical and effective meta-training framework for improving learned optimizers' generalization to longer training horizons.

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

## A  APPENDIX

**To be added!**

