# OpenReview forum: "Towards Efficient Unroll Generalization in Learned Optimizers"
_ICLR.cc/2026/Conference — ICLR 2026 Conference Withdrawn Submission_

### Official Review · Reviewer_3mNj · 2025-10-16

**Soundness:** 2
**Presentation:** 2
**Contribution:** 2
**Rating:** 4
**Confidence:** 3

**Summary:**

Learned optimizers have shown promise, but due to computational constraints and error compounding, they are meta-trained on short horizons and thus struggle on long horizons that may appear during meta-testing. This paper proposes ELO, a method to improve the generalization ability of learned optimizers to horizon lengths longer than what is usually seen during meta-training. It combines two ideas from the reinforcement learning literature, a replay buffer and behavior cloning, to stabilize meta-training and improve generalization. The replay buffer allows restarting from previously trained meta-optimizers, effectively increasing the horizon length without increasing the computational burden. Behavior cloning enforces that the learned optimizer is similar to Adam at the beginning of meta-training, thereby improving stability. Experiments are done on ImageNet-1K (32x32) and FineWeb-10B illustrating the efficacy of the method over AdamW and the original learned optimizer method, and ablations on the main components are also done.

**Strengths:**

- The paper is clear and easy to understand. The motivation and the algorithm are explained clearly, and the figures are illustrative.
- The algorithm itself is straightforward to understand, combining two well-established techniques from reinforcement learning.
- The topic of learned optimizers is an important one, having the potential to improve training of a wide range of models.

**Weaknesses:**

I believe that the main weakness of this paper lies in its experimental set-up.
- First, ELO is quite similar to the algorithm in Chen et al. (2020). Chen et al. (2020) uses a curriculum of horizon lengths (which is a baseline in the experiments) along with off-policy imitation learning instead of behavior cloning. Therefore, I think that it should be included as a baseline in the experiments as well.
- Only one meta-optimizer architecture and size were used in the experiments. It would be helpful to do experiments on a range of architectures or sizes, in order to see the applicability and scability of ELO.

A second minor weakness is novelty. ELO provides a novel combination of experience replay and behavior cloning to the learned optimizer literature, but it is quite similar to the algorithm in Chen et al. (2020) as stated above. Thus, it would be helpful if the paper further discussed the relation of ELO to the algorithm in Chen et al. (2020). The appendix is also missing.

**Questions:**

- Why was the size of the buffer chosen to be 4? Isn't that a bit small?
- Wouldn't it be more fair to have Adam as a baseline instead of AdamW, since Adam is used as the expert in ELO?
- Why is the accuracy so low in Table 1? Is it because of the size of the optimizer architecture?
- Can ELO handle OOD tasks with architectures with different numbers of parameters, or do they have the same number of parameters (lines 361-369)?

---

> ### Author Response · Authors · 2025-12-03
>
> Thank you for the constructive suggestions. For questions about the novelty of our approach and its differences from Chen et al. (2020), please refer to Point 4 in the main comment.  For the concern regarding the accuracies reported in Table 1, please see Point 5 in the main comment.

---

### Official Review · Reviewer_mhUx · 2025-10-26

**Soundness:** 2
**Presentation:** 2
**Contribution:** 2
**Rating:** 2
**Confidence:** 4

**Summary:**

This paper tackles a critical and well-known challenge in the field of learned optimizers (LOs): their poor generalization to long optimization horizons ("unroll generalization"). The authors propose ELO (Efficient Long-horizon Learning), a meta-training scheme designed to address this issue through two key components:

1. ELO uses a replay buffer to store and sample intermediate training checkpoints. This allows new unrolls to start from an advanced state (e.g., step K) instead of always from step 0. This mechanism cleverly reallocates computation from repeatedly re-training the "easy" early steps to exploring longer, more challenging horizons, thereby achieving efficient long-unroll training without extra linear cost.

2. The authors astutely observe that naively jumping into long unrolls (as "Buffer_only" in Fig 2) is unstable due to compounding errors. To solve this, ELO introduces an online behavior cloning (BC) curriculum. The meta-loss is a scheduled blend of a task loss and a BC loss, which forces the LO to imitate a stable expert (Adam). This stabilizes the volatile early stages of meta-training.

The empirical results, while limited by the experimental setup, demonstrate that ELO-LO outperforms naive LOs, curriculum-based LOs, and AdamW on the specific tasks chosen, particularly in the long-horizon regime (10,000 steps).

**Strengths:**

1. The core contribution—using a replay buffer not for its traditional sample efficiency but to efficiently reallocate compute to longer unroll horizons—is a insightful concept.

2. Unroll generalization is arguably one of the biggest hurdles preventing the practical, widespread adoption of LOs. This paper directly attacks this fundamental problem.

3. The solution is a necessary combination of two components. The ablation studies (Fig 2 and Fig 7) convincingly show that both the Replay Buffer and the Behavior Cloning are essential. The buffer-only approach fails, proving that the BC component is a crucial stabilizer.

**Weaknesses:**

1. The credibility of the paper's claims is severely undermined by the choice of meta-training and evaluation datasets. Using ImageNet-1K at a 32x32 resolution (Sec 5.1) is unreasonable. This low resolution discards the fine-grained details that make ImageNet a challenging task. While resource constraints are understandable, conclusions about generalization drawn from this setup are not convincing for practical, large-scale scenarios (which typically use 224x224 or 256x256 resolutions).

2. The paper cites relevant, state-of-the-art LOs like VeLO (Metz et al., 2022), which also claim generalization to 10k+ steps. However, the paper fails to include these in the experimental comparison. The baselines are limited to AdamW and two simplistic LO variants (naive, curriculum), which is insufficient to demonstrate that ELO is a competitive, state-of-the-art approach to unroll generalization.

3. The method's stability seems heavily tied to the quality of the expert (Adam). This reliance is problematic, as the authors admit (Sec 6) that using "more advanced" experts like AdamW was "challenging" and "did not yet observe improvements." Furthermore, using an expert optimizer for behavior cloning is not a new contribution in itself, such as VeLO use the value output of hand-crafted optimizer as part of the input for guidance, making the novelty of this component limited.

4. The replay buffer is central to the "Efficient" claim, yet it seems underdeveloped. In Sec 4.1, the default size is set to just 4. This is an exceptionally small number for a buffer and is not well-justified. The impact of buffer size and sampling strategy is not explored.

5. The paper placeholder "A APPENDIX - To be added!" is a major omission.

**Questions:**

see Weakness

---

> ### Author Response · Authors · 2025-12-03
>
> We greatly appreciate your valuable advice.

---

### Official Review · Reviewer_P3hF · 2025-10-28

**Soundness:** 3
**Presentation:** 1
**Contribution:** 2
**Rating:** 2
**Confidence:** 4

**Summary:**

This paper proposes ELO, a novel learned optimizer (LO) meta-training algorithm that mitigates the out-of-distribution generalization issue of LOs when applied to optimize for a much longer horizon compared to the training distribution. The is claimed that ELO integrates behavior cloning and replay buffer. Experiments show ELO generalizes well on out-of-distribution optimization horizons compared to baselines, and consistently outperforms AdamW. A detailed ablation study is provided.

**Strengths:**

The algorithm proposed is novel and inspiring.

The central claim is well supported by the experiments.

**Weaknesses:**

Need references to support the limitations of existing LOs and many other statements.

There should be space before citations.

Should use \citet for Chen in line 088-089

Line 135-136 needs to spell rms out.

Should make it clear that the parameters of the optimizee are randomly initialized in lines 158-160.

Line 202-203 needs a citation or evidence.

Need to state what assumptions equation (7) requires.

I believe the loss defined in equation (9) does not match the standard behavior cloning, which optimizes the difference between policy actions and expert actions, not the difference between states.

Overall, the writing needs to be significantly improved.

If I'm correct, ELO-LO is meta-trained on unroll length beyond 1000 steps or even 10000 steps, although each meta-training iteration only unrolls 1000 steps. Therefore, the claim that Section 5.4 tests on unrolls beyond the meta-training horizon is wrong.

Similarly, I disagree that the memory buffer used in ELO is a replay buffer in the sense of RL.

**Questions:**

Why is Figure 2 an example of unroll sampling?

Is the \alpha_t in equation (8) the same as that in equation (10)?

Is there a relation between \zeta and \pi_\zeta?

How is LoL-LO different from ELO-LO?

---

> ### Author Response · Authors · 2025-12-03
>
> Thank you so much for the detailed suggestions. Regarding the definition of the BC loss, please refer to Point 3 in the main comment. For the questions “Is the \alpha_t in Equation (8) the same as that in Equation (10)?” and “Is there a relation between \zeta and \pi_\zeta?” please see Points 7 and 8 in the main comment.

---

### Official Review · Reviewer_xJF6 · 2025-10-30

**Soundness:** 1
**Presentation:** 3
**Contribution:** 2
**Rating:** 2
**Confidence:** 4

**Summary:**

Learned optimizers (LOs) can outperform hand-designed ones but struggle to generalize over long unrolls. Simply extending unrolls is costly and unstable. The proposed Efficient Long-horizon Learning (ELO) uses a replay buffer and online behavior cloning to improve long-unroll stability and efficiency.

**Strengths:**

1. The paper addresses a critical and significant challenge, ‘improving unroll generalization’ in the LO field. Solving this problem is a key bottleneck for making LOs practical for real-world tasks.

2. The proposed ELO scheme is a novel combination of two complementary ideas.

3. The replay buffer is a wise idea to recycle computation and expose the LO to longer effective horizons without the additional cost of back-propagation. The online Behavior Cloning (BC) component is a reasonable way to stabilize training.

**Weaknesses:**

1. [Lack of Evidence on Efficiency]
  The paper's central claims of ‘efficiency’ and operating ‘without adding extra meta-training cost’ are unsubstantiated and appear to be incorrect.
  ELO requires calculating both the LO's update and the expert's (Adam) update at every inner step, plus the BC loss. This is a significant computational overhead compared to a naive LO.
  The replay buffer in the paper must store four full copies of the model parameters (theta) and the expert's optimizer states. This means a massive VRAM overhead, combined with the need for the expert's optimizer states for the BC.
  Also, the paper provides no empirical data (e.g., wall-clock time, peak VRAM usage) to support its efficiency claims against baselines.

2. [Questionable Practical Value]
  The empirical gains over the simple and strong AdamW baseline are quite marginal. Given the significant compute and memory overheads (from W1), the paper does not make a compelling case for why a user would use this complex method over AdamW for such a small benefit.

3. [Unclear Contribution of Proposed Methods]
  The paper's narrative focuses on the replay buffer as the main contribution. However, the ablation in Figure 7 shows that Behavior Cloning (LO-BC) is responsible for the vast majority of the performance gain. The step from LO-BC to ELO-LO (adding the buffer) provides very little extra benefit. This suggests the paper's main contribution is actually the online BC technique, which is framed as just a ‘stabilizer’.

4. [Lack of Analysis on Expert (Adam) Choices]
  The method's success depends on using Adam as the expert. The authors briefly state that AdamW was not ‘reliable’, but provide no analysis as to why. Why does the AdamW fail as an expert, while Adam succeeds?
  It also creates an inconsistency by using Adam as a teacher but AdamW as the main baseline in Figure 3-5.

5. [Limited Scale of Experiments]
  All experiments use very small-scale models and low-resolution data. The authors acknowledge this limitation. It is completely unproven whether these results will scale to modern, large-scale models (e.g., LLMs) or hold for truly long-term (e.g., 1M) training.

**Questions:**

See the weakness part.

---

> ### Author Response · Authors · 2025-12-03
>
> Thank you very much for your thoughtful feedback. For weaknesses 2 and 3, please refer to our responses to Points 1 and 2 in the main comments.

---

### Author Response · Authors · 2025-12-03
**Author reply to the main reviewer concerns**

We would like to thank all reviewers for taking the time to read and evaluate our paper. We are pleased to hear that reviewers xJF6 and P3hF​​ recognize the novelty of our work, that xJF6 and mhUx believe we address a fundamental problem for L2O, and that 3mNj believes our paper is clear and easy to understand. After careful consideration, we have decided to withdraw our submission. We now clarify the main concerns of the reviewers.

---
### Clarification of main concerns
&nbsp;

1.&nbsp;&nbsp;**Questionable Practical Value.** First, the expert and the LO share the same set of auxiliary accumulators (e.g., first and second moment), so there is no additional memory overhead. The extra computation introduced by the expert is also negligible compared with the overall training cost. Besides, Figures 3, 4, and 5 show that ELO delivers substantial improvements over the best AdamW baseline on most tasks. We expect this advantage to become even more pronounced as we further scale up the meta-training regime.

2.&nbsp;&nbsp;**Unclear contribution of the buffer relative to behaviour cloning.** Our primary goal is to enable efficient long unroll meta training. As discussed in the paper, the buffer provides a mechanism for sampling long unrolls efficiently, while the behavior cloning objective stabilizes meta-training under buffer initialization and helps reduce compounding errors. Both components are essential and play complementary roles rather than competing ones. It should be noted that without the buffer, strong performance for long horizon unrolls is not reached (see Figure 7).



3.&nbsp;&nbsp;**loss defined in equation (9) does not match the standard behavior cloning.** Our formulation is actually correct. At each inner step, both the learned optimizer and Adam predict the next optimizee state from the same current state. Because of this one-to-one correspondence, computing the loss on the predicted state is equivalent to computing it on the update term produced by each optimizer.

4.&nbsp;&nbsp;**Comparison with Chen et al. (2020).** Chen et al. (2020) proposed two techniques for building general-purpose L2O with strong overall performance:

&nbsp;&nbsp;&nbsp;&nbsp;**(1) Curriculum Learning for L2O:** a training algorithm where the unroll length of the inner problem is progressively increased until the validation loss computed on the latest optimizer parameters stops decreasing.

&nbsp;&nbsp;&nbsp;&nbsp;**(2) Behaviour cloning for L2O:** An alternating optimization approach where the learned optimizer randomly alternates between offline behaviour cloning on a hand-designed optimizer trajectories and optimizing the sum of per-timestep losses.

In contrast, ELO specifically targets efficient long-unroll meta-training. ELO uses a replay buffer instead of a curriculum as a mechanism for extending the duration of inner-problems; ELO simultaneously trains on online behavior cloning and per-timestep loss objectives instead of alternating between them; and, unlike Chen et al. (2020), ELO decays the importance of the behavior cloning objective as training progresses to encourage the optimizer to learn better solutions than the expert. It also should be noted that the naive curriculum approach in Figures 3,4,5 is a version of Chen et al. (2020) algorithm (1). In the next iteration of our paper, we will also compare with Chen et al. (2020)’s method.


5.&nbsp;&nbsp;**Low accuracies in Table 1.** The accuracies are low because we intentionally used a downsampled 32x32 version of ImageNet for simplicity.

6.&nbsp;&nbsp;**General improvements to the paper.** In the future, we plan to further refine the writing and presentation of the paper and broaden our experimental evaluation.


---
### Clarification of the writing
&nbsp;

7.&nbsp;&nbsp;**Whether \alpha_t in Equation (8) is the same as in Equation (10).** Yes, they refer to the same quantity.

8.&nbsp;&nbsp;**Relation between \zeta and \pi_\zeta.** \zeta indicates the auxiliary accumulators (e.g. momentum, step) of the inner state S, and  \pi_\zeta indicates extracting the step index from S.

---

### Note · Authors · 2025-12-04

I have read and agree with the venue's withdrawal policy on behalf of myself and my co-authors.